

# Does the combination of exercise and cognitive training improve working memory in older adults? A systematic review and meta-analysis

Yiqing Wu[1], Ming Zang[2], Biye Wang[1,3] and Wei Guo[1,3]

[1] College of Physical Education, Yangzhou University, Yangzhou, China
[2] College of Electrical Engineering, Chuzhou Polytechnic, Chuzhou, China
[3] Institute of Sports, Exercise and Brain, Yangzhou University, Yangzhou, China

## ABSTRACT

**Background.** Cognitive functioning is dependent on working memory and a decline in working memory is the main cause of cognitive aging. Many studies have suggested that physical exercise or cognitive intervention can effectively improve working memory in the elderly. However, it is still unknown whether a combination of exercise and cognitive training (CECT) is more effective than either intervention alone. The present systematic review and meta-analysis were undertaken to evaluate the effect of CECT on working memory in the elderly.

**Methods.** The review was registered in the International Prospective Systematic Review (PROSPERO, CRD42021290138). Systematic searches were conducted on Web of Science, Elsevier Science, PubMed and Google Scholar. The data were extracted according to the PICOS framework. Comprehensive meta-analysis (CMA) software was used to perform the meta-analysis, moderator analysis and publication bias testing.

**Results.** The current meta-analysis included 21 randomized controlled trials (RCT). Results showed that CECT had a significantly greater impact on working memory in older adults compared to no intervention groups (SMD = 0.29, 95% CI [0.14–0.44], $p < 0.01$), with no significant difference between CECT and exercise (SMD = 0.16, 95% CI [$-0.04$–0.35], $p = 0.12$) or cognitive intervention alone (SMD = 0.08, 95% CI [$-0.13$–0.30], $p = 0.44$). Furthermore, the positive effect of CECT was moderated by intervention frequency and cognitive state.

**Conclusions.** The CECT can effectively improve working memory of older adults, but the effect of CECT compared to single intervention needs to be further explored.

# INTRODUCTION

Working memory (WM), a system of temporarily storing and efficiently manipulating information, is essential to effective cognitive functioning (*Baddeley, 2012*). A decline in working memory is the main cause of cognitive aging, and seriously affects daily life of the elderly (*Goh & Park, 2009*; *Salthouse, 2012*). However, it has recently been

Corresponding author
Wei Guo, guowei@yzu.edu.cn

shown that the aging brain maintains plasticity (*Passow, Thurm & LiC, 2014*). Therefore, interventions that improve working memory in the elderly may delay cognitive aging (*Lampit, Hallock & Valenzuela, 2014*; *Logan, 2014*; *Bonnechère, Langley & Sahakian, 2020*; *Cox & Lautenschlager, 2020*).

For elderly with dementia or other cognitive disorders, non-pharmaceutical methods can reduce medical expenses and improve the quality of the life. Physical exercise is a promising non-pharmacological behavioral intervention for cognitive aging, inhibiting cognitive decline related to age and neurodegenerative diseases (*Bherer, Erickson & Liu-Ambrose, 2013*; *Lin et al., 2019*; *Jia et al., 2019*). Previous meta-analyses have also demonstrated the efficacy of physical exercise training on working memory in the elderly with or without dementia or mild cognitive impairment (MCI), which is a transitional state between normal aging and dementia (*Petersen, 2004*; *Law et al., 2020*; *Zhidong et al., 2021*). One possible mechanism for this effect is the increase in brain-derived neurotrophic factor (BDNF) associated with exercise, as higher expression levels of BDNF correlate with slower cognitive decline (*Buchman et al., 2016*).

In addition to physical exercise, cognitive training intervention has been demonstrated to effectively delay cognitive aging (*Nguyen, Murphy & Andrews, 2019*). Cognitive training may enhance dopaminergic neurotransmission, leading to the consolidation and enhancement of cognition performance in older adults (*Passow, Thurm & LiC, 2017*). Prior studies have shown that cognitive training-based interventions have a positive impact on cognitive functioning in the elderly at a variety of cognitive states (*Giuli et al., 2016*). Furthermore, multi domain cognitive training has proven effective in improving working memory in healthy older people (*Hong et al., 2021*).

Over the last decade, research has found that the combination of cognitive training and exercise (CECT) can improve cognitive function to a larger extent than exercise or cognitive interventions alone (*Benzing & Schmidt, 2017*; *Niederer et al., 2019*). The combined intervention is a multimodal intervention strategy, combining physical exercise with cognitively challenging activities. However, the impact of the combination of the two interventions in an older population remains controversial. Some research has found a superior effect of CECT in healthy older adults compared to physical exercise or cognitive training alone (*Shatil, 2013*; *Zhu et al., 2016*; *Joubert & Chainay, 2018*). Similar results were also found in the elderly with MCI, *i.e.,* the combined intervention also have significant cognitive benefits in older adults with MCI (*Köbe et al., 2016*; *Law et al., 2014*; *Shimada et al., 2018*; *Yang et al., 2020*). While some studies have yielded inconsistent results, showing that combined intervention have no advantage compared to single exercise or cognitive training (*Fiatarone Singh et al., 2014*; *Hackney et al., 2015*; *Chainay, Joubert & Massol, 2021*). Thus, no definitive conclusions have been drawn about the superiority of combined exercise and cognitive intervention. In addition, combined intervention can be divided into simultaneous or sequential modes (*Tait et al., 2017*). Sequential mode provides intervention modalities in separate sessions, usually during the same period (*Ngandu et al., 2015*; *Damirchi, Hosseini & Babaei, 2018*; *Fabre et al., 2002*; *Legault et al., 2011*). Simultaneous mode means that cognitive training and exercise are to be conducted at the same time (*e.g.,* dual task, exergames and video dancing) (*Eggenberger et al., 2016*;

*Ordnung et al., 2017*; *Norouzi et al., 2019*; *Adcock et al., 2020*). However, it is still unclear which combination modes is most beneficial for cognitive function in older adults. Therefore, more investigations are needed to demonstrate the benefit of combined exercise and cognitive training.

Previous research on combined interventions has concentrated on the effects of CECT on overall cognition (*Joubert & Chainay, 2018*; *Karssemeijer et al., 2017*; *Lauenroth, Ioannidis & Teichmann, 2016*; *Zhu et al., 2016*) or executive functioning (*Guo et al., 2020*; *Wollesen et al., 2020*) and few studies focused on working memory. To our knowledge, there have been no previous meta-analyses performed to study the effect of CECT on working memory in older adults. Thus, the main aim of this meta-analysis was to investigate whether CECT is effective in delaying the decline of working memory in older adults.

# MATERIALS & METHODS

This meta-analysis was conducted according to the Preferred Reporting Items for Systematic Reviews and Meta-Analyses (PRISMA) statement. The protocol of this review was registered in the International Prospective Systematic Review (PROSPERO, CRD42021290138).

## Search strategy

Electronic searches of online databases including Web of Science, Elsevier Science, PubMed and Google Scholar were conducted to collect relevant literature since the start of each database until April of 2021. The following keywords were used in the searches: exercise intervention terms ("aerobic exercise" OR "multidomain exercise training" OR "physical activity") AND cognitive intervention terms ("mental training" OR "cognitive training" OR "working memory training") AND combination of intervention terms ("combined" OR "simultaneous" OR "exergame" OR "dual-task" OR "video dancing" OR "cognitive-motor game") AND relevant working memory terms ("cognitive function" OR "working memory" OR "executive function") AND older adults terms ("old" OR "elderly" OR "aging"). Additional articles were obtained through a list of references from published reviews.

## Inclusion criteria

Inclusion and exclusion criteria were determined based on PICOS (population, intervention, comparison, outcome, and study design). The following criteria were used for study inclusion: (1) population: participants were elderly (aged 60 and over) and included healthy individuals, along with those with MCI and dementia; (2) intervention: combination of exercise and cognitive trainings; (3) comparison: at least one comparison group (combined intervention group with cognitive intervention alone or physical exercise intervention alone or a control group receiving no intervention); (4) outcome: measurements of working memory included at least one outcome that could be used to calculate an effect size; (5) study design: randomized controlled trial (RCT) design; and (6) other: written in English. Exclusion criteria included: (1) population: participants aged less than 60 years old; (2) intervention: without combination of exercise and cognitive intervention; (3) comparison: studies that did not compare the combined intervention

group with other intervention groups alone; (4) outcome: studies without relevant data on working memory pre and post intervention; (5) study design: no RCT design; and (6) others: noninterventional studies, reviews and theoretical articles, case and protocol articles, unpublished studies and papers, studies that were not written in English.

## Data extraction and assessment of study quality

In this systematic review, working memory tasks were included based on following criteria: (1) participants are required to store and manipulate information during the task, such as Digit Span Backward test, N-back task, and Corsi block-tapping task, (2) the authors declared the tasks were used to measure working memory (*You et al., 2009*; *Damirchi, Hosseini & Babaei, 2018*). Tasks that simply stored information with little manipulation were excluded, such as Short-term Memory task, Immediate Recall, Forward Digit Span, the Word List test (*Linde & Alfermann, 2014*; *Mrakic-Sposta et al., 2018*; *Romera-Liebana et al., 2018*; *Jardim et al., 2021*).

Data referring to working memory tasks were extracted from combined intervention *versus* exercise intervention alone, or cognitive intervention alone, and no intervention using a spreadsheet. The mean, standard deviation (SD), and the number of participants (N) in each group at pretest and posttest were extracted. If the means and SDs were not mentioned, the change values of the means and SDs after intervention or mean difference of the 95% confidence interval were extracted.

To assess study quality, two independent researchers evaluated the risk of bias for individual studies using the PEDro scale (*Maher et al., 2003*). Each criterion met on the scale was scored a point, for a maximum score of 11. Articles scoring 8 or higher were considered high quality while those scoring below 8 which were considered low quality.

The moderator analysis was conducted to assess the effects of combined intervention on working memory in the elderly. The following moderators were analyzed: the mode of combination (separate *versus* simultaneous), cognitive status (healthy *versus* MCI *versus* dementia), intervention length (short *versus* medium *versus* long), frequency (<3 sessions per week *versus* $\geq$ 3 sessions per week), session length ($\leq$ 60 min *versus* >60 min), no intervention group (active *versus* passive), study quality (high quality *versus* low quality).

## Data analysis

The statistical data was quantified using comprehensive meta-analysis (CMA) software and calculated the effect size. The effect of the intervention was measured by the standardized mean difference (SMD) of changes after the combined intervention group *versus* the no intervention group, single exercise intervention group and single cognitive intervention group (*Borenstein et al., 2009*). The calculation equation of SMD are as follows:

$$SMD = \frac{\overline{x}_1 - \overline{x}_2}{S_{within}}$$

$$S_{within} = \sqrt{\frac{(n_1 - 1)S_1^2 + (n_2 - 1)S_2^2}{n_1 + n_2 - 2}}.$$

If one study had multiple task that measured working memory, outcomes were combined into an average effect size to avoid influencing results. The combined effect size with 95% confidence interval (CI) was calculated to determine the efficacy of CECT on working memory. The $I^2$ index was calculated to analyze the heterogeneity of the included studies. 0% indicated that no heterogeneity has been observed and the higher of the $I^2$ value reflected on more significant of the heterogeneity. The Egger's regression intercepts and funnel charts were performed to estimate publication bias. Funnel charts was symmetrical which indicated no risk of bias and studies were outside the funnel sharp reflecting on high risk of bias. If $I^2 < 50\%$, $p \geq 0.05$, representing the studies had no statistical heterogeneity, so the fixed effect model was used for analysis and if $I^2 \geq 50\%$, $p \geq 0.05$, indicating had statistical heterogeneity between the studies, so the random effect model was used for analysis.

## RESULTS

### Identification of studies

Preliminarily, a total of 1,379 articles were obtained from database searches, of these, 275 articles were removed for duplication and 1,104 were excluded on the basis of titles or abstracts. Of the remaining 55 articles, 38 were excluded based on information found in the full text. Finally, 21 eligible articles were included in this meta-analysis, four of which were included from previous reviews. The specific process of article selection, according to the recommended PRISMA flow diagram is shown in Fig. 1.

### Characteristics of included studies

Twenty-one RCT articles were included in the current meta-analysis. The characteristics of included studies are shown in Table 1. Among the included studies, 16 were conducted on healthy older adults (*Fabre et al., 2002*; *You et al., 2009*; *Legault et al., 2011*; *Maillot, Perrot & Hartley, 2012*; *Shatil, 2013*; *Gschwind et al., 2015*; *Nishiguchi et al., 2015*; *Rahe et al., 2015*; *Eggenberger et al., 2016*; *Schättin et al., 2016*; *Ordnung et al., 2017*; *Kalbe et al., 2018*; *Norouzi et al., 2019*; *Adcock et al., 2020*; *Takeuchi et al., 2020*; *Dana, 2019*), four focused on older adults with MCI (*Suzuki et al., 2013*; *Combourieu Donnezan et al., 2018*; *Damirchi, Hosseini & Babaei, 2018*; *Bae et al., 2019*), and one focused on older adults with dementia (*Karssemeijer et al., 2019*). Sample size ranged from 13 to 153. The mean age range for older adults was 65.8 to 79.2. Of all 21 studies, six included four comparison groups, three included three comparison groups, and 12 included two comparison groups. Of the 21 studies, seven studies used separate interventions, while 14 studies adopted a simultaneous combination of exercise and cognitive intervention, (three of the 14 used the dual task and seven of the 14 used the exergame). The length of intervention ranged from four to 24 weeks and the frequency was one to six sessions each week. Five of the studies included active control groups (leisure activity and health education) and the rest included passive control groups (adhered to normal life). Types of exercise interventions included aerobic training, flexibility training, muscle strength training, balance training, endurance training, and resistance training. The types of cognitive interventions included multidomain cognitive training, working memory training, and cognitive games. Only five

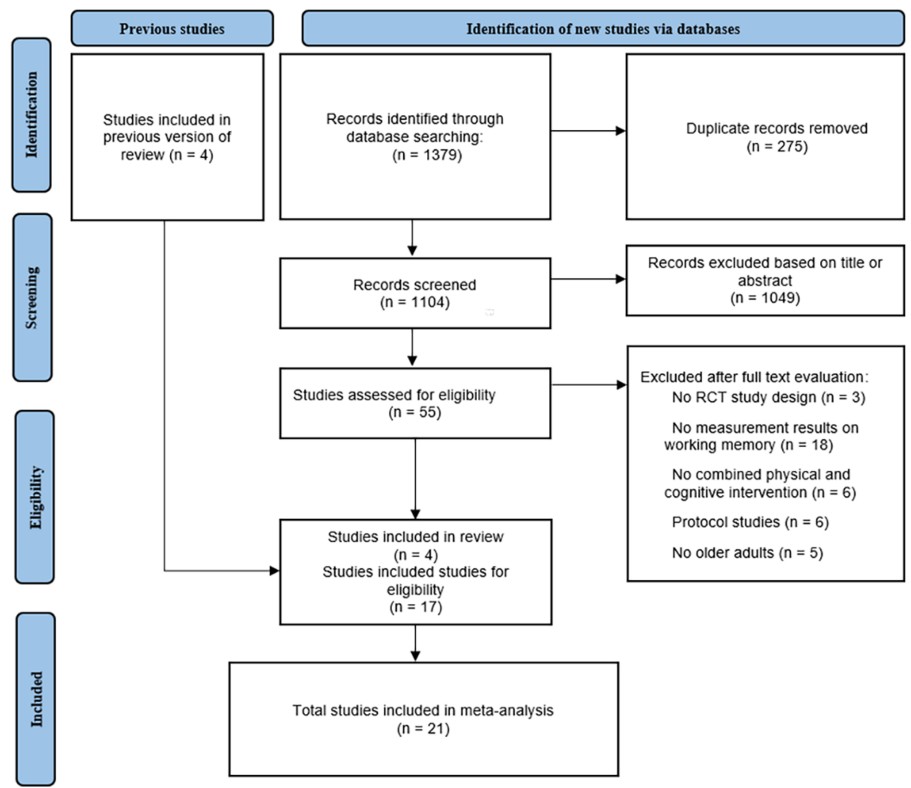

**Figure 1** PRISMA flow chart of study selection process.

articles (*Maillot, Perrot & Hartley, 2012*; *Suzuki et al., 2013*; *Ordnung et al., 2017*; *Damirchi, Hosseini & Babaei, 2018*; *Norouzi et al., 2019*) explicitly reported the gender distribution of participants, of which there were a total of 134 male and 101 female participants. The other included articles did not clearly report the gender distribution of participants. The quality of studies ranged from six to 11 on the PEDro scale and the specific scoring details are shown in Table S1.

### Combined intervention *versus* no intervention group

Figure S1 presented the funnel plot of included studies and the result showed no significant asymmetry (Egger's regression intercept = 1.50, $p > 0.05$).

A total of 15 articles (*Adcock et al., 2020*; *Bae et al., 2019*; *Combourieu Donnezan et al., 2018*; *Damirchi, Hosseini & Babaei, 2018*; *Dana, 2019*; *Fabre et al., 2002*; *Gschwind et al., 2015*; *Karssemeijer et al., 2019*; *Legault et al., 2011*; *Maillot, Perrot & Hartley, 2012*; *Nishiguchi et al., 2015*; *Norouzi et al., 2019*; *Ordnung et al., 2017*; *Shatil, 2013*; *Suzuki et al., 2013*) reported on the effects of CECT compared to no intervention on working memory in older adults. A fixed effect model for meta-analysis was used, as the heterogeneity test showed $Q_{(14)} = 17.31$, $p = 0.24$, $i^2 = 19.10$. As shown in Fig. 2, the combined intervention group showed significant improvement in working memory compared to the no intervention group (SMD = 0.29, 95% CI [0.14–0.44], $p < 0.01$).

**Table 1  Characteristics of included studies.**

| Study | Sample size | Age (Mean) | Cognitive status | Comparison | Cognitive intervention | Exercise intervention | Combination mode | Intervention plan | WM measures task | No intervention group activities | Study quality |
|---|---|---|---|---|---|---|---|---|---|---|---|
| Fabre 2002 | 32 | 60–76 (65.9) | Health | CECT:CT:ET:CG | Mental training | Aerobic training | Separate | 60-90 min/session, 3 sessions/week, 8 weeks | WMS-LM I | Leisure activities | 7 |
| You 2009 | 13 | 64–84 (68.3) | Health | CECT:ET | Memory recall tasks | Walking | Simultaneous (dual-task) | 30 min/session, 5 sessions/week, 6 weeks | Memory recall task | No | 6 |
| Legault 2011 | 67 | 70–85 (76.4) | Health | CECT:CT:ET:CG | Memory training | Aerobic, flexibility training | Separate | 50–150 min/session, 3.5 sessions/week, 16 weeks | The 1-Back, 2-Back Tests | Health Education | 8 |
| Maillot 2012 | 30 | 65–78 (73.5) | Health | CECT:CG | Nintendo Wii games | Nintendo Wii games | Simultaneous (exergame) | 60 min/session, 2 sessions/week, 12 weeks | Spatial Span task | No | 7 |
| Suzuki 2012 | 50 | 65–92 (76.0) | MCI | CECT:CG | Cognitive stimulation | A multicomponent exercise | Simultaneous | 90 min/session, 2 sessions/week, 12 weeks | WMS-LM I | Health Education | 7 |
| Shatil 2013 | 122 | 65–93 (76.8) | Health | CECT:CT:ET:CG | Multidomain cognitive training | Multidomain exercise training | Separate | 40-45 min/session, 6 sessions/week, 16 weeks | Auditory working memory test | No | 6 |
| Nishiguchi 2015 | 48 | ≥60 (73.3) | Health | CECT:CG | Cognitive-motor training | Multidomain exercise training | Simultaneous | 90 min/session, 1 sessions/week, 12 weeks | WMS-LM I ,1-back | No | 8 |
| Rahe 2015 | 68 | 50–85 (68.4) | Health | CECT:CT | Multidomain cognitive training | Strength, endurance, flexibility, coordination | Separate | 90 min/session, 2 sessions/week, 7 weeks | WAIS-II(DSB) | No | 8 |
| Gschwind 2015 | 153 | ≥65 (74.7) | Health | CECT:CG | iStoppFalls exergame with additional cognitive tasks | Balance, strength | Simultaneous (exergame) | 60 min/session, 3 sessions/week, 16 weeks | Digit Span Backward (DSB) | Education booklet (passive) | 8 |
| Eggenberger 2016 | 33 | >65 74.9 | Health | CECT:ET | Cognitive-motor training | Cognitive-motor training | Simultaneous (exergame) | 30 min/session, 3 sessions/week, 8 weeks | working memory test | No | 8 |
| Schättin 2016 | 27 | ≥65 79.2 | Health | CECT:ET | Cognitive-motor training | Cognitive-motor training | Simultaneous (exergame) | 30 min/session, 3 sessions/week, 8-10 weeks | Working memory test | No | 8 |
| Ordnung 2017 | 30 | 69.2 | Health | CECT:CG | The Microsoft X box 360 ™ | The Microsoft X box 360 ™ | Simultaneous (exergame) | 60 min/session, 2 sessions/week, 6 weeks | n-back task | No | 7 |
| Damirchi 2017 | 44 | 60–85 (68.4) | MCI | CECT:CT:ET:CG | "Modified My Better Mind" program | Walking | Separate | 45 min/session, 3 sessions/week, 8 weeks | Forward Digit Span test | No | 7 |
| Kalbe 2018 | 55 | 50–85 (68.1) | Health | CECT:CT | Multidomain cognitive training | Strength, flexibility, coordination, and endurance | Separate | 90 min/session, 2 sessions/week, 7 weeks | Digit Span Backward (DSB) | No | 8 |
| Donnezan 2018 | 69 | 76.7 | MCI | CECT:CT:ET:CG | Multidomain cognitive training | Aerobic training on bikes | Simultaneous | 60 min/session, 2 sessions/week, 12 weeks | the Digit Span Backward test | No | 7 |
| Bae 2019 | 83 | ≥65 (75.9) | MCI | CECT:CG | "KENKOJISEICHI"system | "KENKOJISEICHI"system | Simultaneous | 90 min/session, 2 sessions/week, 24 weeks | Corsi block-tapping task | Health Education | 8 |
| Norouzi 2019 | 60 | ≥65 (68.3) | Health | CECT:ET:CG | 12 cognitive tasks | Resistance training | Simultaneous (dual-task) | 60–80 min/session, 3 sessions/week, 4 weeks | n-back | No | 7 |

Wu et al. (2023), *PeerJ*, DOI 10.7717/peerj.15108

**Table 1** (*continued*)

| | Characteristics | | | | Intervention methods | | | | WM measures task | No intervention group activities | Study quality |
|---|---|---|---|---|---|---|---|---|---|---|---|
| Study | Sample size | Age (Mean) | Cognitive status | Comparison | Cognitive intervention | Exercise intervention | Combination mode | Intervention plan | | | |
| Karssemeijer 2019 | 115 | >60 (79.2) | Dementia | CECT:ET:CG | Multidomain cognitive training | Aerobic bicycle training | Simultaneous (exergame) | 30–50 min/session, 3 sessions/week, 12 weeks | WAIS-III Digit Span WMS-III Spatial Span | Relaxation and flexibility exercises | 8 |
| Dana 2019 | 48 | 60–75 (65.8) | Health | CECT:CT:ET:CG | Multidomain cognitive training | Aerobic treadmill exercises | Separate | 60 min/session, 3 sessions/week, 8 weeks | Working memory test | in the waiting list | 7 |
| Adcock 2020 | 31 | 65–70 (73.9) | Health | CECT:CG | The Active@Home exergame | The Active@Home exergame | Simultaneous (exergame) | 30–40 min/session, 3 sessions/week, 16 weeks | The digit span backward task | No | 8 |
| Takeuchi 2020 | 93 | 65–75 (65.9) | Health | CECT:CT:ET | Working memory training | Aerobic exercise training | Simultaneous (dual-task) | 60 min/session, 3 sessions/week, 12 weeks | A digit span task, WMS-LM I, 2 back | No | 8 |

**Notes.**

MCI, mild cognitive impairment; CECT, combination of exercise and cognitive training; ET, exercise training; CT, cognitive training; CG, control group (no intervention group); WMS-LM I, Wechsler Memory Scale-Logical memory immediate recall; DSB, Digit Span Backward; WAIS-III, Wechsler Adult Intelligence Scale-III.

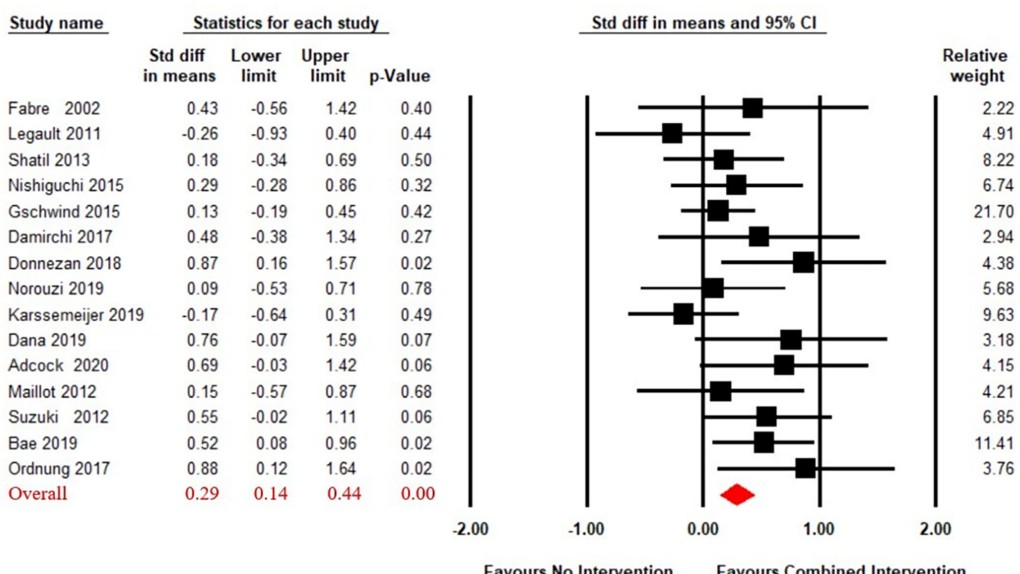

**Figure 2** Forest plot for the effect sizes of the combined interventions compared to the no intervention.

## Combined intervention *versus* single exercise intervention

The funnel plot revealed that one study had a disproportionately large effect size and Egger's test was significant (Egger's regression intercept = 2.71, $p < 0.05$), so one study was removed from further analysis (*Norouzi et al., 2019*). The funnel plot after removal of one outlier was presented in Fig. S2 and had no significant asymmetry.

Eleven studies (*Combourieu Donnezan et al., 2018*; *Damirchi, Hosseini & Babaei, 2018*; *Dana, 2019*; *Eggenberger et al., 2016*; *Fabre et al., 2002*; *Karssemeijer et al., 2019*; *Legault et al., 2011*; *Schättin et al., 2016*; *Shatil, 2013*; *Takeuchi et al., 2020*; *You et al., 2009*) compared the effect of CECT to single physical exercise. The results of our analysis revealed no significant difference on working memory in the effect of between CECT and single physical exercise (SMD = 0.16, 95% CI [−0.04–0.35], $p = 0.12$) (Fig. 3). The heterogeneity test was not significant ($Q_{(10)} = 8.46$, $i^2 = 0$, $p = 0.58$), so a fixed effect model was used.

## Combined intervention *versus* cognitive intervention alone

The funnel plot showed that the result had no significant asymmetry (Fig. S3). Nine articles (*Fabre et al., 2002*; *Legault et al., 2011*; *Shatil, 2013*; *Rahe et al., 2015*; *Combourieu Donnezan et al., 2018*; *Damirchi, Hosseini & Babaei, 2018*; *Kalbe et al., 2018*; *Dana, 2019*; *Takeuchi et al., 2020*) compared the effect of combined intervention to cognitive intervention alone. The results (Fig. 4) of our meta-analysis revealed no significant difference in the effects of combined intervention *versus* cognitive intervention alone on working memory (SMD = 0.08, 95% CI [−0.13–0.30], $p = 0.44$). The heterogeneity test was not significant ($Q_{(8)} = 2.13$, $i^2 = 0$, $p = 0.98$), so a fixed effect model was used.

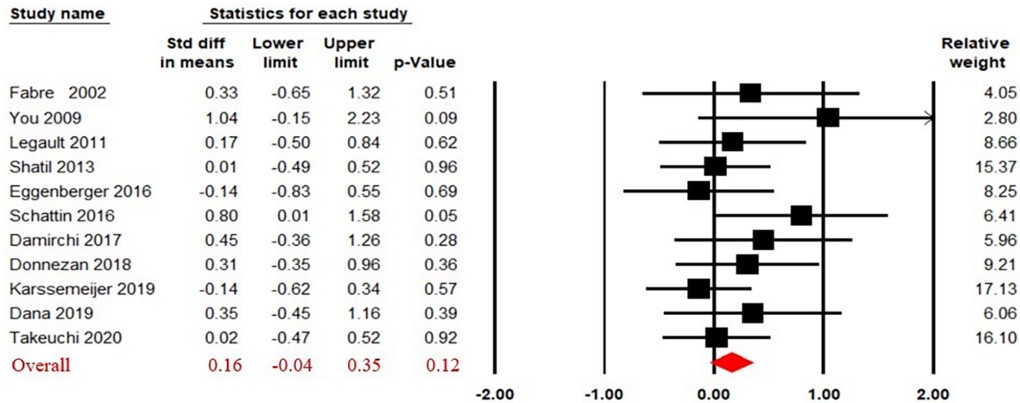

**Figure 3** Forest plot for the effect sizes of the combined interventions compared to the exercise intervention.

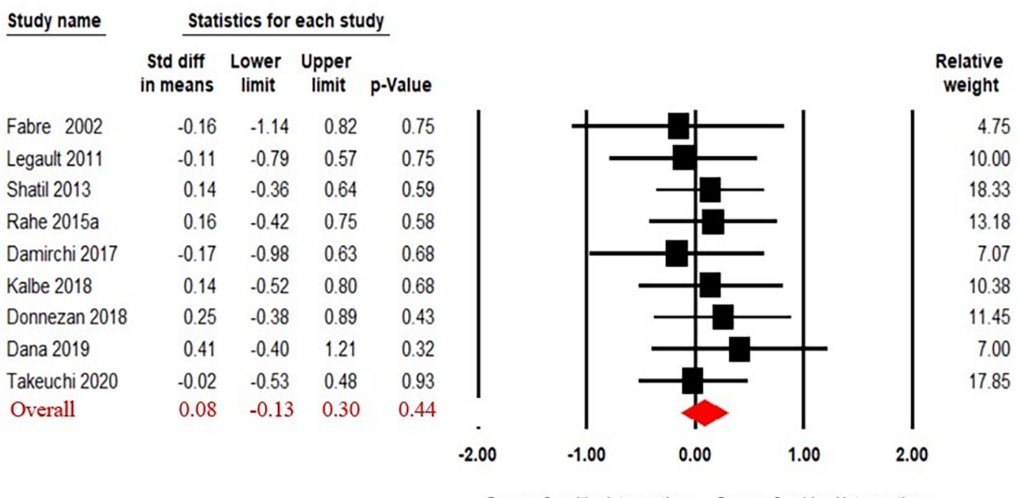

**Figure 4** Forest plot for the effect sizes of the combined interventions compared to the cognitive intervention.

## Moderator analysis

The results of the moderator analysis are shown in Table 2. A heterogeneity test revealed a significant influence of cognitive status on the effects of CECT on working memory ($Q_{(2)} = 7.67$, $p = 0.02$). The criterion used to categorize participants' cognitive status was based on explicit declarations made by the authors, identifying individuals as either having mild cognitive impairment (MCI) or dementia. Combined intervention had a significantly influence among healthy participants (SMD = 0.24, 95% CI [0.06–0.42], $p = 0.01$) and those with MCI (SMD = 0.58, 95% CI [0.29–0.87], $p < 0.01$) but no significant effect among those with dementia (SMD = −0.17, 95% CI [−0.64–0.31], $p = 0.49$).

**Table 2** Moderator analysis for the combined intervention group *vs.* the no intervention group.

| Moderator | Level | No. of studies | SMD | 95% Confidence interval | $I^2$ | Homogeneity test | | |
|---|---|---|---|---|---|---|---|---|
| | | | | | | Q | df | p |
| Mode of combination | Separate | 5 | 0.23 | −0.09 to 0.55 | 4.25 | 0.17 | 1 | 0.68 |
| | Simultaneous | 10 | 0.31** | 0.14 to 0.47 | 30.56 | | | |
| Cognitive status | Healthy | 10 | 0.24* | 0.06 to 0.42 | 0.00 | 7.67 | 2 | 0.02 |
| | MCI | 4 | 0.58** | 0.29 to 0.87 | 0.00 | | | |
| | Dementia | 1 | −0.17 | −0.64 to 0.31 | 0.00 | | | |
| Intervention length | Short (<12 weeks) | 5 | 0.48* | 0.13 to 0.83 | 0.00 | 1.47 | 2 | 0.48 |
| | Medium (≥12 to <16 weeks) | 5 | 0.27 | 0.01 to 0.53 | 42.92 | | | |
| | Long (≥16 weeks) | 5 | 0.23* | 0.03 to 0.44 | 30.95 | | | |
| Frequency | Low (<3 sessions/week) | 6 | 0.52** | 0.28 to 0.76 | 0.00 | 5.53 | 1 | 0.02 |
| | High (≥3 sessions/week) | 9 | 0.15 | −0.03 to 0.34 | 4.19 | | | |
| Session length | Short (≤60 mim) | 9 | 0.28** | 0.09 to 0.46 | 34.94 | 0.06 | 1 | 0.81 |
| | Long (>60 min) | 6 | 0.31* | 0.07 to 0.55 | 0.00 | | | |
| No intervention group | Active | 5 | 0.22 | −0.03 to 0.47 | 48.93 | 0.44 | 1 | 0.51 |
| | Passive | 10 | 0.33** | 0.14 to 0.51 | 0.38 | | | |
| Study quality | Low quality (<8 scores) | 9 | 0.44** | 0.21 to 0.67 | 0.00 | 2.80 | 1 | 0.09 |
| | High quality (≥8 scores) | 6 | 0.18 | −0.01 to 0.38 | 39.36 | | | |

**Notes.**
*$p < 0.05$.
**$p < 0.01$.

A heterogeneity test revealed a significant influence of intervention frequency on the effects of CECT on working memory ($Q_{(1)} = 5.53$, $p = 0.02$). CECT was significantly effective in low frequency intervention (SMD = 0.52, 95% CI [0.28–0.76], $p < 0.01$), but no significant effect in high frequency intervention (SMD = 0.15, 95% CI [−0.03–0.34], $p = 0.11$).

In terms of mode of combination, simultaneous (SMD = 0.31, 95% CI [0.14–0.47], $p < 0.01$) training were proven effective, but no significance was found for sequential (SMD = 0.23, 95% CI [−0.09–0.55], $p = 0.16$) training. CECT was significantly effective regardless of intervention time: Short (SMD = 0.48, 95% CI [0.13–0.83], $p = 0.01$); medium (SMD = 0.27, 95% CI [0.01–0.53], $p = 0.05$); and long intervention length (SMD = 0.23, 95% CI [0.03–0.44], $p = 0.03$). Combined intervention was significantly effective in low quality (SMD = 0.44, 95% CI [0.21–0.67], $p < 0.01$) studies, and had no significance in high quality (SMD = 0.18, 95% CI [−0.01–0.38], $p > 0.05$) studies. Compared with no intervention, long combined intervention sessions (SMD = 0.31, 95% CI [0.07–0.55], $p = 0.01$) and short combined sessions (SMD = 0.28, 95% CI [0.09–0.46], $p < 0.01$) were both significantly beneficial for working memory. In terms of the no intervention group, CECT had a positive effect in passive (SMD = 0.33, 95% CI [0.14–0.51], $p < 0.01$) but no significant effect in active control group (SMD = 0.22, 95% CI [−0.03–0.47], $p = 0.08$).

## DISCUSSION

The current meta-analysis included 21 articles and explored the effect of CECT on working memory in older adults. The results support efficacy of CECT on working memory as compared to those receiving no intervention and the effect is modulated by the frequency of intervention and cognitive status.

The results indicate that there is a significant benefit of CECT on working memory in the elderly compared to no intervention. However, the effects of combined intervention are not significantly different than either physical exercise or cognitive intervention alone. These results disagree with previous meta-analysis and there are potential reasons for this discrepancy. First, other studies have explored the effect of CECT on cognitive function (*Zhu et al., 2016*; *Gheysen et al., 2018*; *Yang et al., 2020*; *Meng et al., 2021*), while the current study specifically explored working memory. Second, there are a limited number of studies directly comparing efficacy of CECT with single exercise or cognitive intervention. Third, prior studies compared the efficacy of CECT on cognitive function in healthy older adults (*Zhu et al., 2016*), while the current meta-analysis extended findings to those with MCI and dementia. Lastly, different types of exercise intervention (*e.g.*, aerobic exercise, flexibility, balance, coordination, and endurance training) may affect cognitive functioning differently.

A combination of systematic exercise training and stimulation of cognitive tasks not only promotes new connections between nerve cells in the brain, but also promotes the formation of new neurons and enhances synapses and plasticity (*Bennett et al., 1964*). The CECT is effective for improved cognitive function and delayed cognitive decline, with positive effects for older people to participate in activities of daily living and improved quality of life.

The current meta-analysis showed that the effects of CECT on working memory in the elderly is moderated by the frequency of intervention and cognitive state. CECT showed significant positive improvements in working memory in both healthy individuals and those with MCI. These results are in line with previous studies showing that CECT improves overall cognitive function and executive function (*Gavelin et al., 2020*; *Guo et al., 2020*). In the current study, low frequency interventions were significantly beneficial to working memory comparing to high frequency interventions. While there is general consensus that greater training frequency generates greater benefits, the results of the current meta-analysis found that low frequency intervention is more than three times as effective as high frequency intervention. Some previous studies have also indicated that high-frequency interventions are less effective than low-frequency interventions (*Lampit, Hallock & Valenzuela, 2014*; *Guo et al., 2020*) suggesting that high-frequency interventions may cause cognitive fatigue in older people (*Holtzer et al., 2011*), negatively affecting cognitive functioning. Moreover, although several studies have indicated that CECT can have a positive impact on the cognitive function of older adults, there is still controversy surrounding which combination mode is more effective. Some meta-analyses have reported that both simultaneous and sequential CECT can improve the cognitive function of older adults (*Zhu et al., 2016*; *Guo et al., 2020*). However, a different meta-analysis has found a

significant effect only for simultaneous CECT, with no notable effect for sequential CECT (*Gheysen et al., 2018*). This outcome is consistent with the results of the present meta-analysis. One possible explanation for this finding is the temporary nature of the increase in brain-derived neurotrophic factor (BDNF). Previous research has demonstrated that higher levels of BDNF can enhance working memory performance (*Erickson et al., 2013*); however, BDNF levels typically return to baseline within 10–60 min after physical activity (*Knaepen et al., 2012*). Thus, simultaneous CECT may offer the greatest neurotrophic benefits compared to sequential CECT.

It is important to note that there was no significant difference between CECT and single physical exercise or cognitive intervention. These results may be affected by the low number of studies that have compared CECT with single physical exercise or cognitive intervention, as majority of studies compare combined intervention to no intervention. This is a notable limitation of the current study. Briefly, the efficacy of CECT was demonstrated, but whether CECT is more beneficial than either physical exercise or cognitive intervention alone is not clear. In the future, it is advisable that studies include multiple intervention groups for comparison. In addition, the majority of current studies on the efficacy of CECT have used some cognitive tasks, such as Digit Span task, Trail Making Test, Stroop task and so on. Future studies are needed to investigate whether biological risk factors (*e.g.*, vascular risk factors and APOE genotype) would influence the efficacy of CECT, which could increase the level of evidence. Furthermore, given the substantial number of studies included in the current meta-analysis utilizing exergame interventions, it may be worthwhile for future studies to investigate the impact of such interventions on working memory in older adults.

The current research has some other limitations that should be noted when considering the results. First, the current study did not investigate post-intervention follow-up and therefore, the maintenance of improved working memory as a result of intervention cannot be assessed. Second, working memory is a complex cognitive domain and was measured in a variety of ways in the included studies, possibly introducing variability within the results. Finally, other potential moderator variables, such as baseline levels of exercise or cognitive health, were not analyzed.

## CONCLUSIONS

The presented meta-analysis indicates that CECT is a promising way to improve working memory in older adults and that the effect is influenced by the frequency of intervention and cognitive status. However, there is no evidence that combined intervention is superior to exercise or cognitive intervention alone.

### Funding

This work was supported by the National Social Science Foundation of China (Grant No. 18CTY014). The funders had no role in study design, data collection and analysis, decision to publish, or preparation of the manuscript.

## Grant Disclosures

The following grant information was disclosed by the authors:
The National Social Science Foundation of China: 18CTY014.

## Competing Interests

The authors declare there are no competing interests.

## Author Contributions

- Yiqing Wu performed the experiments, analyzed the data, prepared figures and/or tables, authored or reviewed drafts of the article, and approved the final draft.
- Ming Zang performed the experiments, analyzed the data, prepared figures and/or tables, and approved the final draft.
- Biye Wang conceived and designed the experiments, analyzed the data, authored or reviewed drafts of the article, and approved the final draft.
- Wei Guo conceived and designed the experiments, analyzed the data, authored or reviewed drafts of the article, and approved the final draft.

## Data Availability

 The raw measurements are available in the Supplemental Files.

## Supplemental Information

Supplemental information for this article can be found online at http://dx.doi.org/10.7717/peerj.15108#supplemental-information.

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
