# Peer review of "Does the combination of exercise and cognitive training improve working memory in older adults? A systematic review and meta-analysis"

_PeerJ, doi:10.7717/peerj.15108_

## Round 0.1 · original submission · Major Revisions

Dear authors,

Please reply point by point to the reviewers' comments.

Reviewer 1 ·

Basic reporting

Dear authors, I’ve read your manuscript, the study is mostly complete. The conclusions drawn are supported by the literature analysis. The manuscript is generally well-written and easy to read; a slight spell-check is required. Although the results of the study are interesting, I have some methodological and other concerns that the authors need to address before acceptance and publication.

The whole manuscript must be justified.

Abstract
Methods section: Please add the PROSPERO registration, remove the keywords used for the search and add how you have extracted the data (PICOs?), risk of bias assessing tool used.

keywords usually should be different from that used in the main title

Introduction
The literature on the subject is sufficiently well summarised.
Line 72: explain, in extenso, MCI: Mild Cognitive Impairment (MCI)

Experimental design

The methods section is sufficiently well described.

The inclusion and exclusion criteria could be clearer if listed in a table. The selection of eligible studies is the cornerstone of systematic reviews. This will determine what the authors want to focus on. The results of systematic reviews are more valuable than other types of reviews because they provide researchers with the best available evidence for a given question. It is strongly recommended that researchers use population, intervention, comparison, outcome and study design (PICOs) framework (I think you have used this framework but, as already said, a table can make it clearer).

You should add references for PEDro scale.

In the registered PROSPERO protocol, in the search section, you also indicate google scholar, while in the article you do not, is there a particular reason?

Validity of the findings

The results and discussion section are quite clear and organised. The parameters considered are well presented. The discussion section could be improved through a more in-dept comparison of the reviewed studies.

Figure 1: you’ve missed the eligibility tab in the flowchart.

Line 179: “Among the included studies, 16 were conducted on healthy older adults, 4 focused on older adults with MCI, and 1 focused on older adults with dementia”, which ones? Please add relative citations.

Table 1: it might be helpful to indicate the sample composition of the studies (gender of participants)

Reviewer 2 ·

Basic reporting

Profesionally structured and data and figures provided.

1.- Your Abstract may benefit from additional data at line 31 supporting the statement. Additionally, lines 84-85 and 124 could be improved due to unclear writing/use of English.

2.- Table 1 could have been clearer to read if divided into two separate tables.

Experimental design

Concerning to the inclusion of studies:

1.- Please clarify whether a criterion/test was considered for mild cognitive impairment (MCI) and dementia categorisation of the individuals and if so, consider reflecting this information in the relevant section.

2.- Please justify the inclusion of You (2009) and Shatil (2013) considering their higher risk of bias compared to the other studies.

Validity of the findings

Robust and statisticaly sound data were provided.

1.- Considering the high volume of studies including exergame (7 out of the 14 adopting combined exercise-cognitive training), it would have been interesting to determine the effects on working memory solely based on exergame interventions.

Additional comments

This meta-analysis highlights some interesting points about the frequency of the intervention and the cognitive status of working memory in the elderly population. It is a novel approach that considers individuals with MCI and dementia compared to previous studies (Zhidong, 2021; Teixeira-Santos, 2019). However, additional information about the inclusion of MCI and dementia subjects may enhance the validity of the findings. I praise the authors for the data set and figures provided and encourage them to continue their studies in the field.

---

## Round 0.2 · accepted · Accept

Dear authors,

Congratulations!

The manuscript is now ready for publication.

Thank you for choosing PeerJ.

Reviewer 1 ·

Basic reporting

In my opinion, the manuscript is now suitable for publication.

Experimental design

I have no further comments

Validity of the findings

I have no further comments